# Trends of Myocarditis and Endocarditis Cases before, during, and after the First Complete COVID-19-Related Lockdown in 2020 in France

**DOI:** 10.3390/biomedicines10061231

**Published:** 2022-05-25

**Authors:** Thibaut Pommier, Eric Benzenine, Chloé Bernard, Anne-Sophie Mariet, Yannick Béjot, Maurice Giroud, Marie-Catherine Morgant, Eric Steinmetz, Charles Guenancia, Olivier Bouchot, Catherine Quantin

**Affiliations:** 1Department of Cardiology, Dijon University Hospital, 21000 Dijon, France; charles.guenancia@chu-dijon.fr; 2Laboratory of Cerebro-Vascular Pathophysiology and Epidemiology (PEC2) EA 7460, Health Sciences Faculty, University of Burgundy, 21000 Dijon, France; yannick.bejot@chu-dijon.fr (Y.B.); maurice.giroud@chu-dijon.fr (M.G.); 3Biostatistics and Bioinformatics (DIM), Dijon University Hospital, 21000 Dijon, France; eric.benzenine@chu-dijon.fr (E.B.); anne-sophie.mariet@chu-dijon.fr (A.-S.M.); catherine.quantin@chu-dijon.fr (C.Q.); 4Department of Cardiovascular and Thoracic Surgery, Dijon University Hospital, 21000 Dijon, France; chloe.bernard@chu-dijon.fr (C.B.); mariecatherine.morgant@chu-dijon.fr (M.-C.M.); eric.steinmetz@chu-dijon.fr (E.S.); olivier.bouchot@chu-dijon.fr (O.B.); 5Inserm, CIC 1432, Dijon University Hospital, Clinical Investigation Center, Clinical Epidemiology/Clinical Trials Unit, 21000 Dijon, France; 6Department of Neurology, Dijon University Hospital, 21000 Dijon, France; 7High-Dimensional Biostatistics for Drug Safety and Genomics, Paris-Saclay University, UVSQ, Inserm, CESP, 94800 Villejuif, France

**Keywords:** myocarditis, endocarditis, COVID-19, epidemiology

## Abstract

Background. The impact of the COVID-19 pandemic on hospitalization for cardiac infections is not well known. We aimed to evaluate the nationwide trends in hospital stays for myocarditis and endocarditis cases before, during and after the nationwide lockdown for the COVID-19 pandemic in France. We then aimed to describe the proportion of myocarditis and endocarditis patients with and without COVID-19 and their clinical characteristics. Methods. Hospitalized cases of cardiac infection were extracted from the French National Discharge database, which collects the medical records of all patients discharged from all public and private hospitals in France. Age, sex, and available cardiovascular risk factors were compared between stays with and without COVID-19 during the lockdown. Results. The number of myocarditis cases was 11% higher in 2020, compared to the average of the three prior years. In 2020, 439 of 3727 cases of myocarditis were associated with COVID-19. For endocarditis, there was an increase in cases by 7% in 2020 versus prior years. For endocarditis, 3% (240 of 8128 cases) of patients with endocarditis had COVID-19. For myocarditis, older age, hypertension, diabetes, obesity, and atrial fibrillation were more frequent in patients with COVID-19 than in those without. For endocarditis, only hypertension was more frequent in patients with COVID-19 than in those without. Conclusion. Our study reports an increase in hospitalizations for both myocarditis and endocarditis in 2020, possibly related to the COVID-19 pandemic. Interestingly, the trends differ according to the COVID-19 status. Knowledge of the factors associating myocarditis or endocarditis and COVID-19 may improve the quality and the type of monitoring for people with COVID-19, the identification of patients at risk of cardiac infections, and the treatment of COVID-19 patients.

## 1. Introduction

Coronavirus disease 2019 (COVID-19), caused by the severe acute respiratory syndrome coronavirus 2 (SARS-CoV-2), has become a pandemic that continues to cause significant mortality and morbidity worldwide, leading to a public health crisis of unprecedented magnitude. In the absence of treatment or a vaccine, ceasing most human contact was probably the only way to curb the spread of the virus, which is why many countries, including France, imposed a nationwide lockdown during the spring of 2020.

Beyond the direct effects of COVID-19, the pandemic had indirect effects on morbidity and mortality through changes in patient behaviors and the organization of the health system. Indeed, the lockdown significantly changed the hospital landscape during the spring of 2020, with hospital activity almost entirely dedicated to the pandemic and the management of patients with confirmed COVID-19. The reorganization of hospitals led to the cancelation of most scheduled activities in favor of COVID-19 emergency activity, and many patients did not have access to their usual care.

Cardiovascular diseases are a common cause of morbidity and mortality [1,2]. Moreover, in COVID-19, prior cardiovascular disease is a major risk factor for complications and mortality [3,4,5,6]. Therefore, during the pandemic, guidelines advised people with prior cardiovascular disease to be particularly careful about maintaining social distancing measures. Several studies have reported reductions in hospital volumes during the pandemic for cardiovascular diseases such as myocardial infarction [7,8,9,10], heart failure [11,12], or stroke [13,14,15,16,17], which are characterized by high rates of complications and death [3,4].

Myocarditis and endocarditis are not the most common forms of cardiovascular disease, but may have significant morbidity and mortality, particularly if therapeutic management is delayed. Myocarditis remains a challenging diagnosis due to the various clinical presentations: infarct-like syndrome, acute or chronic heart failure, cardiac arrhythmia, or cardiogenic shock. However, chest pain mimicking acute myocardial infarction is the most frequent presentation. Endocarditis is most often characterized as an infectious syndrome with fever, associated with the emergence of valvular heart disease. These symptoms are variable and can be less alarming or sudden than myocardial infarction or stroke, potentially leading to an underestimation of the incidence during the COVID-19 pandemic.

Few data are available on the incidence of myocarditis and endocarditis during the pandemic, either in the general population or in patients with COVID-19 [5,6,11,12]. The aim of this study was to evaluate the nationwide trends in hospitalized myocarditis and endocarditis cases, before (2017 to 2019), during (17 March to 11 May 2020), and after (11 May to 30 September 2020) the nationwide lockdown for COVID-19 in France. The second objective was to describe and compare the proportion of patients with and without COVID-19 and their clinical characteristics.

## 2. Methods

The nationwide data provided by the French National Hospital Discharge database (PMSI–Programme de Médicalisation des Systèmes d’Information) were transmitted by the national agency for the management of hospitalization data. This process was approved by the National Committee for Data Protection. Transmission of these data to a third party is forbidden. Despite the fact that PMSI data are confidential, they are available for researchers who meet specific criteria, from the Agency for Information on Hospital Care (ATIH–Agence Technique de l’Information sur l’Hospitalisation).

### 2.1. Hospitalization Data

Hospitalization data from 1 January to 30 September, for the years 2017 to 2020, were extracted from the PMSI, which collects the medical records of all patients discharged from all public and private hospitals in France. Hospitalizations included possible hospital transfers.

### 2.2. Myocarditis and Endocarditis

Myocarditis and endocarditis cases were identified according to the International Classification of Diseases—Tenth Revision Codes (ICD-10) using codes I40, I41 for myocarditis and I330, I339 for endocarditis. COVID-19 was identified using specific codes created by the ATIH for this pandemic. Codes were extracted for primary diagnoses but also for associated and secondary diagnoses in order to fully explore the two diseases even when another severe disease became the primary diagnosis. Other extracted variables were age, sex, and available cardiovascular risk factors or diseases (hypertension, diabetes, obesity, and atrial fibrillation). The diagnosis of myocarditis was based on clinical symptoms and was confirmed by cardiac magnetic resonance imaging (CMR), while the diagnosis of endocarditis was confirmed by cardiac echography.

### 2.3. Study Design

We retrospectively analyzed the data for all patients living in France and admitted to public and private hospitals in France for acute myocarditis or endocarditis between January and September 2020. This period included the first peak of the COVID-19 pandemic and the national complete lockdown from 17 March (week 12) to 11 May (week 19) 2020. Hospitalization numbers were compared month by month with the numbers from the same period in 2017 to 2019. Clinical characteristics and in-hospital deaths were compared between stays with and without COVID-19 during the lockdown. This retrospective study had no impact on patient care and all data were anonymous. This study was authorized by the French Data Protection Authority on 3 July 2020 (registration number: DR-2020-250 on 07/03/2020).

### 2.4. Statistical Analysis

Qualitative variables are presented as frequencies (percentages). Quantitative variables are presented as medians and interquartile ranges (IQR). The different variables analyzed in the cohort of hospitalized patients were compared using the Chi^2^ test or the Fisher’s exact test (for qualitative variables) and the median test (for quantitative variables), between patients with and without COVID-19 during the lockdown. The change in the number of stays for each disease in 2020 compared with the mean of 2017 to 2019 by month was plotted as smoothed curves using degree 2 spline functions. The threshold for statistical significance was set at <0.05. All analyses were performed using SAS (SAS Institute Inc., Version 9.4, Cary, NC, USA).

## 3. Results

The results are first presented according to the specific disease (myocarditis or endocarditis), and then they are detailed according to the COVID-19 or non-COVID-19 status of the patient during the hospital stay.

### 3.1. Myocarditis

The overall number of myocarditis cases (Figure 1A) was higher in 2020 (January–September 2020) (3727) when compared with the mean number of cases in 2017–2019 (3345, i.e., an increase of 11.4%) during the entire study period, with a slow decrease from May.

In patients with COVID-19 (Figure 1B), there were more hospitalizations for acute myocarditis during the lockdown only. For patients without COVID-19 (Figure 1C), there was a decrease that began before the lockdown, but there was no rebound in the number of cases after the lockdown. We found that 439/3727 (11.8%) COVID-19 cases occurred in patients with myocarditis in 2020.

### 3.2. Endocarditis

The overall number of endocarditis cases (Figure 2A) was higher in 2020 (January–September 2020) (8128) when compared with the mean number from 2017 to 2019 (7580, i.e., an increase of 7.2%). During the lockdown in 2020, the overall number of hospital stays for endocarditis was slightly inferior in March and April compared with the mean from 2017 to 2019 (1677 versus 1733, i.e., a decrease of 3.2%), but the number of cases surpassed the rates seen in previous years in May until September (Figure 2A) (4591 versus 4289, i.e., an increase of 7.0%).

For endocarditis with COVID-19 (Figure 2B), the maximum number of endocarditis cases was observed during the lockdown. For patients without COVID-19 (Figure 2C), we observed a decrease in hospital stays during the lockdown in March–April (1528 in 2020 versus 1733 for the mean of 2017–2019, i.e., a decrease of 11.8%), followed by a period with higher than average numbers from May to August followed by a slow decrease (4510 in 2020 versus 4289 for the mean of 2017–2019, i.e., an increase of 5.2%). During our analysis period, 240/8128 (3.0%) patients with endocarditis had COVID-19.

### 3.3. Clinical Characteristics of Patients According to COVID-19 Status

Concerning the clinical characteristics of patients with myocarditis according to their COVID-19 status (Table 1), patients with COVID-19 were older (61 vs. 50 years, *p* < 10^−4^), and more likely to have other cardiovascular risk factors such as hypertension (36% vs. 20%, *p* < 10^−4^), diabetes (23% vs. 7%, *p* < 10^−4^), or obesity (18 vs. 10%, *p* = 0.0021). There was also more atrial fibrillation (19% vs. 12%, *p* = 0.0033) among myocarditis patients with COVID-19, and five times more deaths in this same group (24% vs. 5%, *p* < 10^−4^).

For endocarditis, we did not observe a significant difference in age between the two groups. In contrast with myocarditis, patients presenting endocarditis with COVID-19 did not have more cardiovascular risk factors than patients without COVID-19, except for hypertension (53% vs. 43%, *p* = 0.024). Mortality was twice as high in endocarditis patients with COVID-19 compared to that in the non-COVID-19 group (32% versus 16%, *p* < 10^−4^).

## 4. Discussion

To the best of our knowledge, this is the first nationwide, population-based retrospective cohort study to compare (according to COVID-19 status) the number and the clinical characteristics of patients hospitalized for myocarditis and endocarditis in all public and private hospitals before, during, and after the lockdown in 2020. This study reports specific trends for these two conditions.

For myocarditis, we observed an increase in hospital stays not only during but also after the lockdown compared with the mean from 2017 to 2019. As expected, the number of hospitalizations for myocarditis in patients with COVID-19 was higher during the lockdown (which corresponded approximately to the first wave), but only during this period. On the contrary, the number of hospitalizations for myocarditis in patients without COVID-19 decreased sharply during the lockdown, followed by an approximate return to the levels observed in 2017–2019, but without surpassing them. For endocarditis, we observed a decrease at the beginning of the lockdown followed by an excessive increase after the end of the lockdown. As with myocarditis, the number of hospitalizations for endocarditis in patients with COVID-19 was higher during the lockdown. For patients without COVID-19, the number of hospitalizations for endocarditis decreased sharply during the lockdown, but this decrease was followed by an excessive rise in cases after the lockdown (exceeding the levels of previous years).

To understand the observed trends, explanations may be provided by the pathophysiology of COVID-19, myocarditis, and endocarditis. The significant rise in hospitalization stays for myocarditis associated with COVID-19 could be explained by the positive role of two clinical presentations leading to the diagnosis of myocarditis, such as fever present in myocarditis or febrile heart failure or chest pain [18,19,20], quickly revealing myocarditis by through CMR. This observation highlights the importance of clinical features in the management of emergencies during the COVID-19 pandemic. However, data on the prevalence of myocarditis associated with COVID-19 may vary since they depend on the methods used. Among 39 patients who died from COVID-19 and were autopsied [21], 24 (61%) presented anatomical lesions of myocarditis, characterized by the three classical anatomical features: inflammation of the myocardium, microvascular thrombosis, and myocardial necrosis [5,22,23]. In living patients, the diagnosis of myocarditis was based on clinical assessment or on biopsy [24,25,26,27,28], resulting in a prevalence between 5% [22] and 27.8% [29,30]. When CMR is used, the data are more informative [30]. On 100 patients with COVID-19 [31], a CMR was performed 70 days after the acute state of the infection, revealing that 78% of the cases had detectable high-sensitivity troponin, 60% had evidence of active myocardial inflammation from abnormal native T1 and T2, 32% manifested late gadolinium enhancement, and 22% had a pericardial involvement. In another study including 145 competitive student athletes aged between 17 and 23 years old [32], a CMR performed 15 days after the diagnosis of mild to moderate COVID-19 revealed myocarditis in two athletes (1.4%). Therefore, as described by Zheng and Thakkar [5,33], COVID-19 may have a direct impact on the myocardium [5,33]. This is suggested by the high prevalence of COVID-19 in our study (11%), compared with the prevalence of 3% in our endocarditis cohort or 6% in myocardial infarction [7] and stroke [13]. Its entry into the myocardium may be boosted via the angiotensin-converting enzyme 2 receptors, which are expressed mainly in the heart and lungs [33,34]. Besides the direct impact of COVID-19 on the myocardium, another mechanism is possible via the systemic release of various pro-inflammatory cytokines such as interleukine-1 and 6, interferon gamma, macrophage inflammatory protein-1A, and tumor necrosis factor-alpha [35]. Inflammatory infiltration is reported to be more frequent than direct damage by the virus [36,37,38]. SARS-CoV-2 gains entry into human cells by binding its spike protein to the membrane protein angiotensin-converting enzyme 2 (ACE2). As previously described, ACE2 can be found on the epithelial cells of the respiratory tract, pneumocytes, and cardiomyocytes. Therefore, it is possible that SARS-CoV-2 infects the human heart, especially in cases of heart failure, because ACE2 is upregulated, although the presence of viral receptors does not always predict tropism. Intracellular SARS-CoV-2 might impair stress granule formation through its accessory protein. Without the stress granules, the virus is able to replicate and damage the cells. Naïve T lymphocytes can be primed for viral antigens with antigen-presenting cells, while cardiotropism was linked by the heart-produced HGF (hepatocyte growth factor, which is an antifibrotic factor). The HGF binds c-Met, an HGF receptor on T lymphocytes. The primed CD8+ T lymphocytes migrate to the cardiomyocytes and cause myocardial inflammation through cell-mediated cytotoxicity. Then, inflammation increases T-lymphocyte activation and releases more cytokines. This results in a positive feedback loop of immune activation and myocardial damage such as myocarditis [39].

Concerning endocarditis, cardiac valve damage could be linked, through the cytokine storm, with the systemic inflammation and hypercoagulable state induced by COVID-19. The formation of vegetations is initiated through bacteremia, which causes microorganisms to adhere to the previously damaged endothelium. The upregulated coagulation state caused by a recent COVID-19 infection also helps microorganisms to become encased in a platelet-fibrin matrix on the heart valve structure [40].

The role of the inflammatory reaction is reinforced by the recent reports of myocarditis after vaccination against COVID-19 in young patients. In this predominantly male population, 148 cases were detected in the first month after vaccination and mainly after the second dose. Considering myocarditis without COVID-19, the deep drop observed during the lockdown is similar to that observed in other cardiovascular diseases such as myocardial infarction [7,8,9,10], heart failure [12], or stroke [13,14,16,41], potentially for the same reasons. For example, many authors have underlined the influence of several factors during the first lockdown: the fear of putting additional pressure on hospitals and caregivers, the messages relayed by the media and health authorities to stay home to avoid the risk of COVID-19 contamination [14,17,18], the priority given to patients with COVID-19, and patient isolation.

In our study, we were able to consider a longer post-lockdown period than in the previous literature. We were, thus, able to show that the recovery in the number of myocarditis cases without COVID-19 did not exceed the number of hospitalizations in previous years. This may be due to several mechanisms, for example a decrease in the number of hospitalized COVID-19 patients, an appropriate reactivation of emergency networks, or changes in patient behavior when they experienced temperature or dyspnea. Two other potential reasons for the recovery of cases to the levels of previous years are delays in echocardiography acts, which were delayed until after the end of the lockdown and the end of the first COVID-19 wave, or the onset of chronic heart failure. The consequence of COVID-19-related myocarditis could explain the onset of chronic heart failure and COVID-19-related chronic fatigue syndrome [42,43,44].

For endocarditis, the rise in the number of hospital stays during the lockdown in patients with COVID-19 appears to be independent from COVID-19. The two conditions were associated in only 3% of the cohort, and endocarditis is not involved in the inflammatory storm, which acts to promotes endothelial activation, leading to a proadhesive and prothrombotic status [23,33]. In fact, we first observed a deep drop during the lockdown, which may have been for the same reasons described above for myocarditis. However, contrary to myocarditis, this deep drop was followed by an increase in the number of hospital stays compared with 2017 to 2019, exceeding the numbers recorded in the post-lockdown period, and then dropping back to previous levels in August. The reasons could be related to a change in medical practices with the activation of the echocardiography networks able to provide a fast evaluation of patients with fever mimicking COVID-19, inflammatory syndrome, or febrile heart failure. In any case, the reason for the increase in the number of cases needs further investigation.

Regarding the clinical characteristics and outcomes associated with COVID-19, it is interesting to observe that a high proportion of the patients with both myocarditis and COVID-19 had several cardiovascular risk factors such as older age, male sex, hypertension, diabetes, obesity, and atrial fibrillation, which was also the case for myocardial infarction [7,8,9,10], heart failure [11,12], and stroke [13,14,16,17]. A high mortality rate was confirmed in the two conditions studied here, which was also the case for the other cardiovascular diseases related to COVID-19 complications [13,19,20,23]. This observation can likely be explained by the fact that patients hospitalized for COVID-19 during the lockdown in 2020 were in more serious condition and had more comorbidities than those who were hospitalized without COVID-19. In addition, COVID-19 infection increased the mortality rate of these patients for non-cardiological reasons, such as respiratory failure or thrombotic complications, and patients without associated COVID-19 infection probably remained free of these complications.

Moreover, Block et al. [45] recently conducted a similar study on the incidence of inflammatory cardiac diseases in the United States. The authors calculated the incidence of cardiac outcomes (myocarditis and or pericarditis) among 15,215,178 individuals aged ≥5 years who had SARS-CoV-2 infection or after mRNA COVID-19 vaccines, stratified by sex (male or female) and age group (5–11, 12–17, 18–29, and ≥30 years). The incidences of inflammatory cardiac complications after SARS-CoV-2 infection or mRNA COVID-19 vaccination were low overall but were higher after infection than after vaccination for both males and females in all age groups.

Limitations: This nationwide study is a descriptive, retrospective, observational study. Possible misclassification may have occurred. However, while under-detection of cases is of course possible, this risk should be greatly reduced by the long post-lockdown period studied. This prolonged period allowed for almost complete capture of the discharge abstracts, which are an obligation for all public and private hospitals. Moreover, we have no data on the causes of endocarditis or the number of surgical cases. Access times to hospital and procedures were not ascertained in discharge abstracts.

Strengths: All French hospital data from public and private hospitals and primary, secondary, and tertiary cardiac units were collected, and we were, thus, able to be nationally representative. The post-lockdown period was evaluated, and the possible risk of a seasonal effect was limited by using data from 2017 to 2019.

## 5. Conclusions

This study reports, for the first time, a specific phenomenon that has not been observed in other cardiac conditions, with a higher number of stays for both myocarditis and endocarditis during the first complete lockdown, in particular, in patients with COVID-19, compared to the average from the three prior years. This observation underlies a potential link between cardiac infections and COVID-19. In contrast, there was a decrease in the hospitalization volume for these two conditions in non-COVID-19 patients during the lockdown that was followed by a significant increase of hospital access after the lockdown, similar to the case for other cardiovascular diseases.

It will be necessary to continue monitoring the trends for these two conditions going forward. These data could be used to ascertain whether the effect was simply a brief recovery in the number of hospitalizations after the end of the lockdown or whether it reflects a deeper, long-lasting shift brought on by changes in the behavior of the general public and health professionals, who may be more sensitive to the risk of cardiac infections than in the past.

## Figures and Tables

**Figure 1 biomedicines-10-01231-f001:**
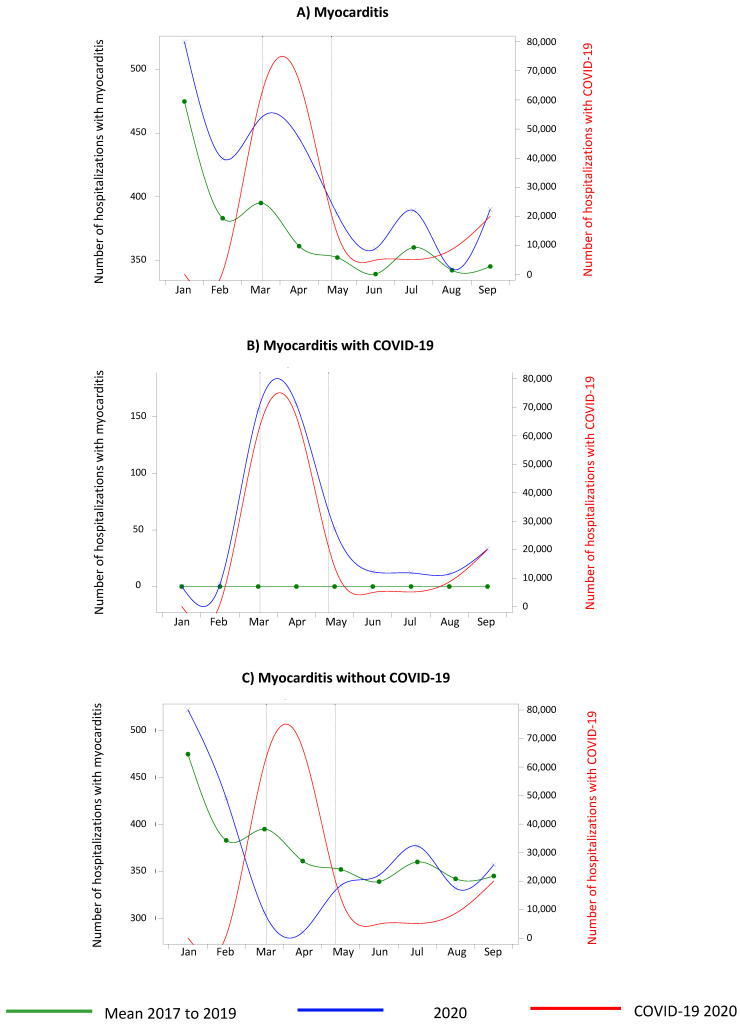
Number of hospitalizations in metropolitan France for myocarditis between January and September 2020 compared with the same months from 2017 to 2019. *Vertical dotted lines indicate the beginning and end of the national lockdown (March 17th and May 10th).* (**A**): Graphical demonstration of the increased number of hospitalizations for myocarditis during lockdown in 2020 in comparison to the mean number of cases from 2017 to 2019. (**B**): Graphical demonstration, for patients with COVID-19, of the increased number of hospitalizations for myocarditis only during lockdown in 2020 in comparison to the mean number of cases from 2017 to 2019. (**C**): Graphical demonstration, for patients without COVID-19, of the decreased number of hospitalizations for myocarditis that began before the lockdown in 2020 and the absence of rebound in the number of cases after the lockdown in comparison to the mean number of cases from 2017 to 2019.

**Figure 2 biomedicines-10-01231-f002:**
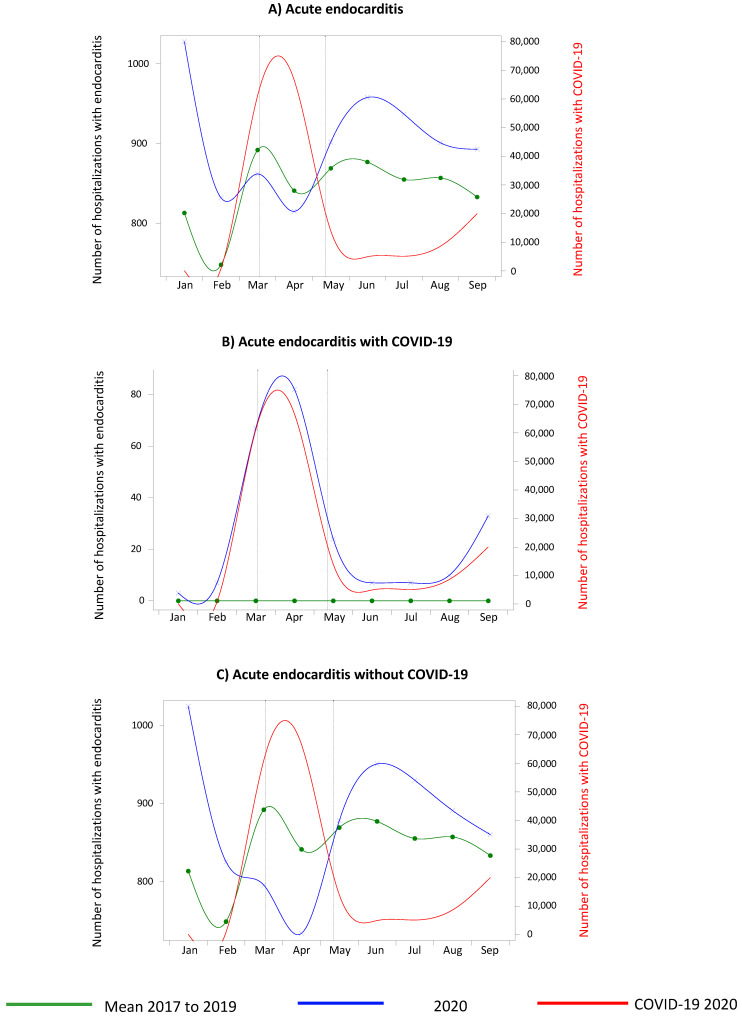
Number of hospitalizations in metropolitan France for acute endocarditis between January and September 2020 compared with the same months from 2017 to 2019. *Vertical dotted lines indicate the beginning and end of the national lockdown (March 17th and May 10th).* (**A**): Graphical demonstration of the decreased number of hospitalizations for acute endocarditis in March and April 2020 and the increased number of these hospitalizations in May until September 2020 in comparison to the mean number of cases from 2017 to 2019. (**B**): Graphical demonstration, for patients with COVID-19, of the increased number of hospitalizations for acute endocarditis during lockdown in 2020 in comparison to the mean number of cases from 2017 to 2019. (**C**): graphical demonstration, for patients without COVID-19, of the decreased number of hospitalizations for acute endocarditis during lockdown in 2020 and the increased number of these hospitalizations after the lockdown in 2020 in comparison to the mean number of cases from 2017 to 2019.

**Table 1 biomedicines-10-01231-t001:** Clinical characteristics of myocarditis and endocarditis among COVID-19 patients vs. non-COVID-19 patients during the lockdown of 2020.

	COVID-19 + N (%)	COVID-19 − N (%)	*p*-Value
**Myocarditis (*n* = 836)**	322	514	
Age, years			
Mean (SD)	61.1 (19.9)	50.2 (20.4)	<10^−4^
Sex			0.0068
Women	104 (32.3)	214 (41.6)	
Men	218 (67.7)	300 (58.4)	
Hypertension	115 (35.7)	101 (19.7)	<10^−4^
Diabetes	75 (23.3)	38 (7.4)	<10^−4^
Obesity	57 (17.7)	53 (10.3)	0.0021
Atrial fibrillation	62 (19.3)	61 (11.9)	0.0033
In-hospital mortality	76 (23.6)	26 (5.1)	<10^−4^
**Endocarditis (*n* = 1473)**	136	1337	
Age, years			
Mean (SD)	69.8 (15.2)	71.0 (14.3)	0.32
Sex			0.079
Women	33 (24.3)	422 (31.6)	
Men	103 (75.7)	915 (68.5)	
Hypertension	72 (52.9)	573 (42.9)	0.024
Diabetes	39 (28.7)	348 (26.0)	0.50
Obesity	21 (15.4)	157 (11.7)	0.21
Atrial fibrillation	43 (31.6)	484 (36.2)	0.29
In-hospital mortality	43 (31.6)	214 (16.0)	<10^−4^

N: number; %: percent; SD: standard deviation.

## Data Availability

The PMSI database was transmitted by the national agency for the management of hospitalization data. The use of these data by our department was approved by the National Committee for data protection. We are not allowed to transmit these data. PMSI data are available for researchers who meet the criteria for access to these French confidential data (this access is submitted to the approval of the National Committee for data protection) from the national agency for the management of hospitalization (ATIH—Agence technique de l’information sur l’hospitalisation). Address: Agence technique de l’information sur l’hospitalisation, 117 boulevard Marius Vivier Merle, 69329 Lyon CEDEX 03.

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
