# Peer review of "Trends of Myocarditis and Endocarditis Cases before, during, and after the First Complete COVID-19-Related Lockdown in 2020 in France"

_biomedicines, 2022, doi:10.3390/biomedicines10061231_

Round 1

Reviewer 1 Report

Comments:

1- There are two Figures 1C, would the authors please check.

2- Line 163: Would the authors please check the data “patients with COVID-19 were older (61 vs 40 years, p<10-4)”. According to Table 1, it should be “60 vs. 50 years”.

3- For Line 261-267: Would the authors please discuss the possible reasons that the significantly higher in-hospital mortality was observed in COVID-19 patients with myocarditis and endocarditis compared with their counterparts in details. To our knowledge, several variables (hypertension, diabetes, obesity artial fibrillation) have been reported to be independent risk factors for COVID-19 mortality.

Author Response

We thank the Reviewer for giving us an opportunity to improve the content of our manuscript.

1- There are two Figures 1C, would the authors please check.

  • The duplicate was deleted.

2- Line 163: Would the authors please check the data “patients with COVID-19 were older (61 vs 40 years, p<10-4)”. According to Table 1, it should be “60 vs. 50 years”.

  • The sentence has been corrected line 163: “patients with COVID-19 were older (61 vs 50 years, p<10-4)”.

3- For Line 261-267: Would the authors please discuss the possible reasons that the significantly higher in-hospital mortality was observed in COVID-19 patients with myocarditis and endocarditis compared with their counterparts in details. To our knowledge, several variables (hypertension, diabetes, obesity artial fibrillation) have been reported to be independent risk factors for COVID-19 mortality.

  • In our work, patients with inflammatory cardiac disease and COVID-19 infection had a higher mortality rate compared to their counterparts. In our view, this observation can be explained by the fact that the patients hospitalized for COVID-19 during the lockdown in 2020 were more in more serious condition and had more comorbidities than others. In addition, COVID-19 infection increased the mortality rate of these patients for non-cardiological reasons such as respiratory failure or thrombotic complications, and probably the patients without associated COVID-19 infection remained free of these complications. A sentence was added in the text.

Reviewer 2 Report

Thibaut POMMIER and colleagues in  "Trends of myocarditis and endocarditis cases before, during and after the first complete COVID-19-related lockdown in  2020 in France" from nationwide data provided by the French National Hospital Discharge database analisis are very interesting and accurately described with results were obtained is much detailed. Even more interesting is the comparison before pandemia and during  also including endocarditis. At this point, the authors can make a brief mention about the biomolecular mechanisms of the onset of endocarditis and myocarditis from the virus.

Author Response

We thank the Reviewer for giving us an opportunity to improve the content of our manuscript.

Thibaut POMMIER and colleagues in  "Trends of myocarditis and endocarditis cases before, during and after the first complete COVID-19-related lockdown in  2020 in France" from nationwide data provided by the French National Hospital Discharge database analisis are very interesting and accurately described with results were obtained is much detailed. Even more interesting is the comparison before pandemia and during also including endocarditis. At this point, the authors can make a brief mention about the biomolecular mechanisms of the onset of endocarditis and myocarditis from the virus.

  • Myocarditis is an inflammatory disease and the most common cause of myocarditis is viral infections, including COVID-19 infection. Possible pathophysiology of SARS-CoV-2–related myocarditis was described in some papers and a paragraph has been added in the discussion (central role of ACE2 and HGF).
  • Nevertheless, the influence of COVID-19 infection on specific pathogenic mechanisms of infectious endocarditis is not clearly specified in the literature. Cardiac valve damage could be linked, due to the cytokine storm, with the systemic inflammation and the hypercoagulable state induced by COVID-19 infection. The formation of vegetation is initiated through bacteremia, which causes microorganisms to adhere to the previously damaged endothelium. The up-regulated coagulation state by recent COVID-19 infection also helps microorganisms to encase in a fibrino-platelet matrix on the heart valve structure.

Two paragraphs have been added in the text.

Reviewer 3 Report

Thank you for the opportunity to review the paper by Pommier et al. It is an interesting observation from a large database in France. Although, the authors have delineated between myocarditis and endocarditis before the pandemic and in the year 2020, the findings of associated risk factors who developed myocarditis are not new. However, there are some novel features in endocarditis, which is with publishing.

Specific comments: Authors should cite ( Block et al. Morbidity and Mortality Weekley report from CDC posted on April 1, 2022, and include it in the discussion to compare the french data with that of the US.

Table-1: legend should change to myocarditis and endocarditis rather than cardiac infections as myocarditis can occur due to secondary effects on the myocardium besides direct myocardial infection.

Figure-1 (there is the repetition of C and represented twice).

Author Response

We thank the Reviewer for giving us an opportunity to improve the content of our manuscript.

Specific comments: Authors should cite ( Block et al. Morbidity and Mortality Weekley report from CDC posted on April 1, 2022, and include it in the discussion to compare the french data with that of the US.

  • We agree with the reviewer and the reference is now cited in the article. We described the results and compared them with our observations. A paragraph was added to the discussion.

Table-1: legend should change to myocarditis and endocarditis rather than cardiac infections as myocarditis can occur due to secondary effects on the myocardium besides direct myocardial infection.

  • The legend of the Table 1 has been modified: “Clinical characteristics of myocarditis and endocarditis among COVID-19 patients vs non-COVID-19 patients during the lockdown of 2020”.

Figure-1 (there is the repetition of C and represented twice).

  • The duplicate was deleted.

Round 2

Reviewer 1 Report

This manuscript can be accepted in present form.

Author Response

Additional small explanations in the legend under the figures were added.